# Selectively Extracting and Injecting Visual Attributes into Text-to-Image Models

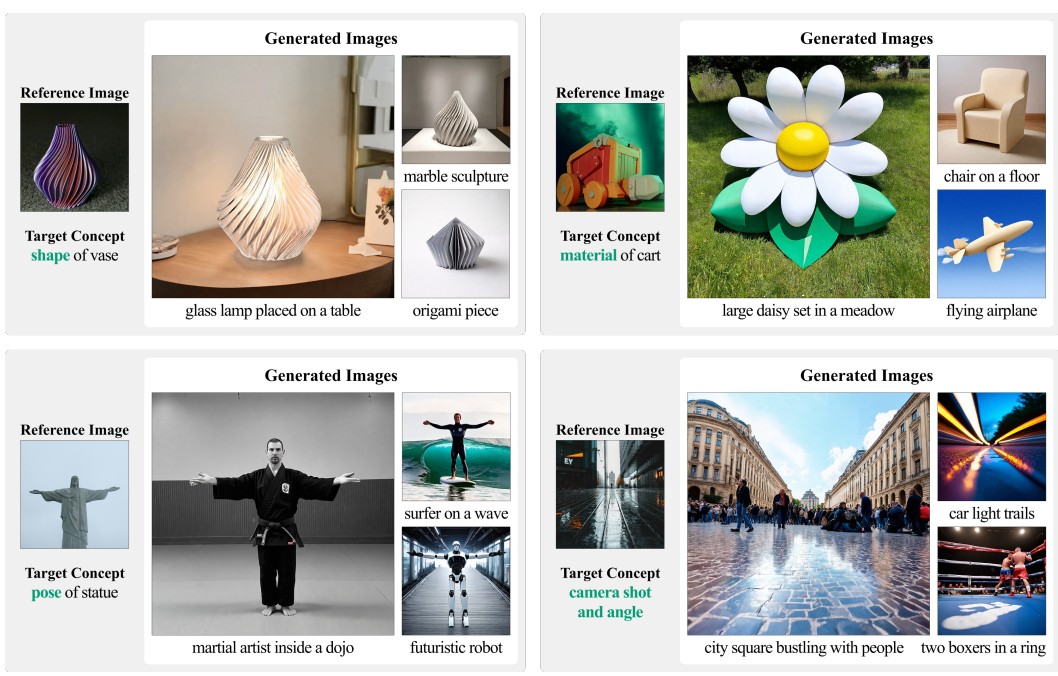

Figure 1: Generation results from the proposed concept learning method. The method selectively extracts an attribute-level concept from a reference image and applies it across diverse contexts.

## Abstract

Text-to-image models are increasingly utilized in design workflows, but articulating nuanced design intentions through text remains a challenge. This work proposes a method that extracts a visual attribute from a reference image and injects it directly into the generation pipeline. The method optimizes a text token to exclusively represent the target attribute using a custom training prompt and two novel embeddings: *distilled embedding* and *residual embedding*. Through this approach, a wide range of attributes can be extracted, including the shape, material, or color of an object, as well as the camera angle of the image. The method is validated on various target attributes and text prompts drawn from a newly constructed dataset. The results show that it outperforms existing approaches in selectively extracting and applying target attributes across diverse contexts. Ultimately, the proposed method enables intuitive and controllable text-to-image generation, streamlining the design process.

## 1 Introduction

Image generation models are increasingly integrated into design workflows (Guo et al., 2024; Barros & Ai, 2024), driven by significant improvements in image quality. These models are typically guided by text prompts (Nichol et al., 2021; Rombach et al., 2022; Ramesh et al., 2022; Saharia et al., 2022;

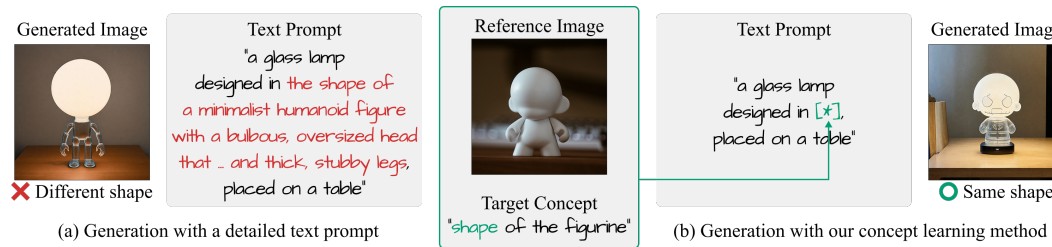

Figure 2: Two text-to-image generation scenarios. (a) A user attempts to reconstruct the target concept with a detailed text prompt, but the generated concept differs. (b) Our method extracts the target concept directly from the reference image and reconstructs the concept successfully.

Betker et al., 2023), generating realistic images that reflect the written descriptions. Designers thus specify shapes, materials, or colors in the prompts to visualize design prototypes.

However, expressing nuanced design intentions solely through text is difficult, which limits the quality of the resulting prototypes. Although designers often collect visual references illustrating particular attributes they wish to adopt (Hassan, 2023), they still spend considerable time experimenting with prompts to reconstruct those attributes. Some even compile "prompt books" by studying prompt-image correlations (OpenArt, 2022), but the generated results deviate from the intended attributes. Fig. 2a illustrates this challenge and highlights the need for a methodology that directly transfers target attributes to text-to-image models.

To address this limitation, variants of text-to-image models that receive image representations along with text have emerged (Mou et al., 2024; Brooks et al., 2023; Gal et al., 2022). However, these models are often restricted to learning only certain types of attributes, such as layouts or styles (Zhang et al., 2023a), or focus on capturing entire subjects (Kumari et al., 2023). Furthermore, many require thousands of training samples and pre-processing to learn a single attribute type.

In this work, we propose a method that enables text-to-image models to learn a wide range of attribute-level concepts from a single reference image. These concepts include the shape, material, or color of an object, as well as broader properties such as style or camera angle. Unlike existing approaches, our method does not require a dataset or pre-processing. As illustrated in Fig. 2b, it extracts the specifications of a target concept directly from an image, effectively bypassing the need for manual prompt engineering.

To extract the target concept, our method optimizes a text token to represent its specifications exclusively. Since multiple attributes are entangled within a single image, it is essential to separate the desired concept from the irrelevant ones. To achieve this, we construct a custom training prompt that roughly separates target from non-target attributes and introduce two novel embeddings: the *distilled embedding* and the *residual embedding*. The distilled embedding robustly removes features associated with non-target attributes, while the residual embedding stabilizes the optimization. Together with the custom training prompt, these embeddings allow the token to be selectively optimized for the target concept. Once optimized, the learned concept can be reconstructed by simply inserting the token into text prompts.

We demonstrate the effectiveness of this method on diverse concepts and prompts drawn from a newly constructed dataset designed specifically for concept learning. The experimental results show that our method outperforms existing approaches in selectively extracting and applying target concepts across diverse contexts. This provides a practical and generalizable solution for integrating visual attributes into text-to-image generation, significantly reducing the manual effort required in design prototyping.

## 2 RELATED WORK

**Image-guided text-to-image generation.** Although text-to-image models (Nichol et al., 2021; Rombach et al., 2022; Ramesh et al., 2022; Saharia et al., 2022; Betker et al., 2023) can already generate high-quality results conditioned on text prompts, efforts have been made to additionally

guide models with image representations for more perceptually intuitive control over the outcome. For example, ControlNet (Zhang et al., 2023a), T2I-Adapter (Mou et al., 2024), and StyleShot (Gao et al., 2024) modify the model architecture to receive representations such as a sketch, pose map, or style image. However, each work only covers a limited range of visual concepts (e.g., layout or style) and requires tons of training data and pre-processing, such as edge detection (Canny, 1986) or pose estimation (Kreiss et al., 2021; Cao et al., 2019), to accept each type of representation. There are image editing models, such as InstructPix2Pix (Brooks et al., 2023) and Emu Edit (Sheynin et al., 2024), that accept unprocessed images, but instructions for editing concepts are only provided in text prompts.

Another line of work is subject-driven generation (Ruiz et al., 2023; Kumari et al., 2023; Kim et al., 2024), which takes a few images of a subject as additional guidance and generates the subject in different environments. This task is generally solved by mapping the images to a token embedding of the text-to-image model through optimization or a separate image encoder. Most studies have focused on parameter-efficient optimization (Kumari et al., 2023; Han et al., 2023), encoder-based learning (Wei et al., 2023; Li et al., 2023a), and multi-subject composition (Avrahami et al., 2023; Ding et al., 2024), starting from the pioneering work, Textual Inversion (Gal et al., 2022). However, these studies are oriented towards learning the subject as a whole and do not provide the freedom to choose which concept to learn. On the other hand, our method can selectively learn only a target concept from a single image.

Separately, a unified text-to-image generation model called OmniGen (Xiao et al., 2024) supports all the aforementioned image-conditioned generation, image editing, and subject-driven generation. For each task, the model demonstrates performance comparable to that of state-of-the-art models. The model also shows generalization ability to unseen tasks, and we evaluate its performance in concept learning.

**Attribute-level concept learning.** The methods for attribute-level concept learning are still in their early stages, with only a few studies attempting to learn concepts beyond subjects by developing Textual Inversion. For example, Vinker et al. (Vinker et al., 2023) proposes a method to decompose a subject learned by Textual Inversion into multiple concepts. This is the first to aim to learn attribute-like concepts, yet the categories of the decomposed concepts are arbitrary. In addition, Huang et al. (Huang et al., 2024) and Motamed et al. (Motamed et al., 2023) develop methods to learn an object relation in reference images or the conceptual difference between pairs of images. However, they only focus on a single category of concepts or require collecting image pairs.

There are also studies that increase the number of token embeddings that are mapped to the reference images. XTI (Voynov et al., 2023) uses different embeddings per layer in the text-to-image model, and ProSpect (Zhang et al., 2023b) uses different embeddings per diffusion timestep. Here, ProSpect demonstrates that embeddings of different timestep possess different attributes of the reference image, such as content, material, style, or layout. The method learns several categories of attribute-level concepts, and we compare our method with ProSpect in the experiments.

## 3 PRELIMINARIES

**Training of diffusion models.** Diffusion models (Sohl-Dickstein et al., 2015; Dhariwal & Nichol, 2021; Rombach et al., 2022) are latent variable models that generate samples from a learned data distribution by iteratively denoising Gaussian noise. Training these models involves predicting the denoised image $\mathbf{x}_0$ from a noisy image $\mathbf{x}_t$ at a timestep $t = 1, ..., T$. In a setting that takes text prompts as input, the objective function of a diffusion model $x_\theta$ is formulated as

$$\mathbb{E}_{\mathbf{x}_0, y, \epsilon \sim \mathcal{N}(\mathbf{0}, \mathbf{I}), t} \| x_\theta(\mathbf{x}_t, t, \tau(y)) - \mathbf{x}_0 \|_2^2, \tag{1}$$

where $\epsilon$ is noise added to $\mathbf{x}_0$ to create $\mathbf{x}_t$, $y$ is a text prompt, and $\tau$ is a text encoder. After training, prediction of the model $x_\theta(\mathbf{x}_t, t, \tau(y))$ enables the generation of a slightly denoised image $\hat{\mathbf{x}}_{t-1}$ from $\mathbf{x}_t$. By iteratively passing the noisy image through the model, starting from $\mathbf{x}_T \sim \mathcal{N}(\mathbf{0}, \mathbf{I})$, a new denoised image $\hat{\mathbf{x}}_0$ that reflects $y$ can be produced.

**Textual Inversion** is a method to inject a unique subject into the output domain of pre-trained text-to-image diffusion models. The method embeds the characteristics of the subject into a token and has the advantage of preserving the parameters (i.e., the manifold) of the models. Specifically,

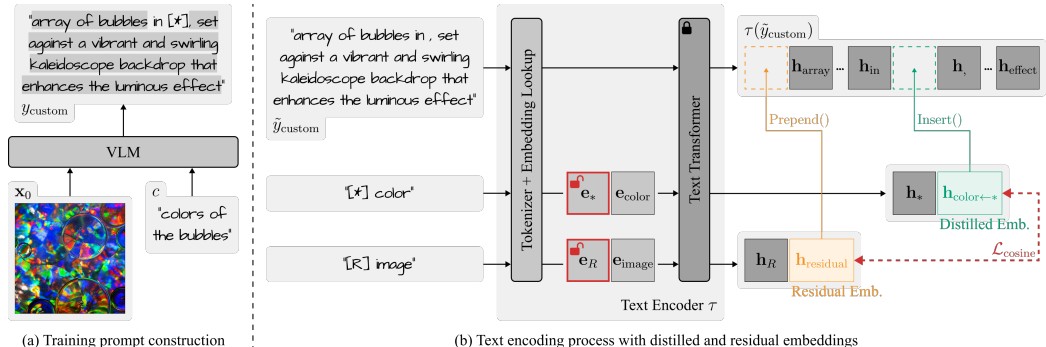

(a) Training prompt construction    (b) Text encoding process with distilled and residual embeddings

Figure 3: An overview of our method.

when several reference images that contain the subject are given, a token embedding $\mathbf{e}_*$ is optimized to minimize the denoising loss of Eq. (1) over those images. The equation of the optimization is written as follows:

$$\mathbf{e}_* = \arg\min_{\mathbf{e}} \mathbb{E}_{\mathbf{x}_0, y, \epsilon \sim \mathcal{N}(\mathbf{0},\mathbf{I}), t} \|x_\theta(\mathbf{x}_t, t, \tau(y)) - \mathbf{x}_0\|_2^2. \quad (2)$$

Here, $\mathbf{x}_0$ is an image sampled from the reference images, and $y$ is a simple training prompt such as "A [*]," where [*] is the placeholder token for the token embedding $\mathbf{e}_*$. Also, the diffusion model $x_\theta$ is frozen during the optimization. After the optimization, the subject is reconstructed by inserting [*] into text prompts (e.g., "A [*] on the moon").

## 4 METHOD

In this work, we build on Textual Inversion and develop a method to learn an attribute-level concept from an image. Our method only requires a single reference image without any dataset or pre-processing of the image.

When a reference image $\mathbf{x}_0$ and a target concept $c$ are given, we optimize a token embedding $\mathbf{e}_*$ to learn the target concept in the image. Here, we assume that $c$ is provided as text, and the concept can be of any category. For example, if we want to learn the colors of the bubbles from $\mathbf{x}_0$, such as in Fig. 3a, we simply set $c$ as "colors of the bubbles."

For the successful learning of an attribute-level concept, it is most important to *avoid learning any untargeted attributes*. Our method is designed with this consideration in mind and proceeds through the following steps. First, we roughly exclude non-target attributes from training by constructing a training prompt that suits our concept learning task (Section 4.1). We leverage the findings in subject-driven generation for the construction. Then, we propose a novel embedding called distilled embedding, which more explicitly excludes the non-target attributes (Section 4.2). Based on the well-known mechanism of text transformers, the distilled embedding robustly isolates the features of the target concept from the optimized token embedding. Lastly, we observe that only using the distilled embedding could destabilize the optimization. Thus, we propose another learnable embedding called residual embedding, and the joint use of the distilled and residual embeddings stabilizes the optimization (Section 4.3).

### 4.1 CONSTRUCTING CUSTOM TRAINING PROMPTS

In Textual Inversion, the optimization loss is minimized when the prompt "A [*]" reconstructs $\mathbf{x}_0$. This means that the method forces the embedding $\mathbf{e}_*$ to represent *all* the attributes of $\mathbf{x}_0$, including objects, backgrounds, and even the camera shot and angle. On the other hand, concept learning requires $\mathbf{e}_*$ to *only* represent the target concept $c$ and avoid capturing the other attributes. This needs our training prompt to be designed such that $\mathbf{x}_0$ can be reconstructed even if $\mathbf{e}_*$ only represents $c$.

Interestingly, a recent study on subject-driven generation has revealed that having a descriptive training prompt helps token embeddings to selectively represent foreground objects (Kim et al., 2024). Specifically, when the training prompt includes descriptions of the backgrounds in the reference

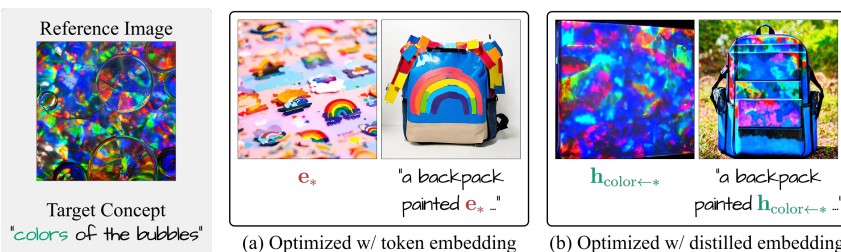

Figure 4: Comparison of the token and distilled embeddings.

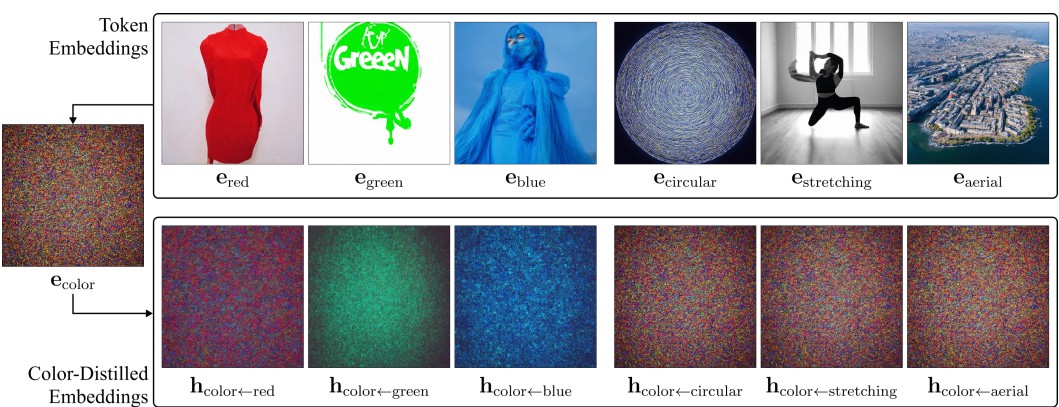

Figure 5: Visualization of token embeddings and distilled embeddings.

images, the embeddings focus on capturing the remaining foregrounds in order to reconstruct the images. This finding is particularly relevant to our concept learning task, and we leverage it to construct our training prompts.

We want the embedding $e_*$ to focus on representing the target concept $c$. Thus, we include descriptions of all the untargeted attributes in $x_0$ in the training prompt. To obtain the descriptions, we utilize a vision-language model (VLM) (Li et al., 2022; 2023b; Wang et al., 2024; Hong et al., 2024). We provide an instruction-tuned VLM with $x_0$ and request that the model describe the image except for $c$ in one sentence. Then, by adding a concept-specific phrase (e.g., "in [*]") to the generated caption, we obtain our custom training prompt $y_{custom}$. We provide the full instructions given to the VLM in the supplementary material.

**Selecting initializer token.** When we optimize $e_*$ to represent $c$, we have to set the starting point of the embedding. An easy way to do this would be to initialize $e_*$ with the category of $c$ (e.g., "color"). However, we can utilize the VLM once again to look for a more suitable initializer token in an automated manner.

After obtaining $y_{custom}$, we ask the VLM to infer what [*] refers to. Based on the model output, we select a few candidate tokens with similar meanings. These candidate tokens are then input back into the VLM, where we prompt the model to choose the most appropriate one. The selected token is used as the initializer token. Details of the candidate selection process and the full VLM instructions are provided in the supplementary material.

### 4.2 DISTILLING TARGET FEATURES FROM TOKEN EMBEDDING

Although our training prompt $y_{custom}$ prevents the token embedding $e_*$ from capturing the other attributes than the target concept $c$, it is impossible to describe all the undesired attributes in the text of limited token length. Thus, some of the untargeted attributes still get embedded in $e_*$. This is evidently shown in Fig. 4a, where the text prompt "[*]" generates not just the color, but also other concepts such as the layout or camera focus.

The root cause of this issue comes from the fact that $\mathbf{e}_*$ can be the token embedding of any value. Since there are no constraints that directly limit the value range of the embedding, any attributes not described in $y_{\text{custom}}$ naturally get embedded in $\mathbf{e}_*$ to lower the optimization loss. While concept-specific phrases such as "in," "made of," or "captured in" indirectly bound the attributes that can go into $\mathbf{e}_*$, it is impossible to structurally prohibit $\mathbf{e}_*$ from learning the undescribed attributes with the current text encoding process.

To prevent any undescribed attributes from being learnt, we propose a novel embedding called *distilled embedding* $\mathbf{h}_{[\text{category}]\leftarrow*}$. Our proposed embedding selectively distills the features that belong to $c$ from the token embedding $\mathbf{e}_*$ through the transformer in the text encoder of the text-to-image model. Specifically, we leverage the well-known mechanism of transformers, where semantically-related tokens in a sentence attend to each other as they pass through the layers (Abnar & Zuidema, 2020). An example of using our distilled embedding is shown in Fig. 4b, which demonstrates that only $c$ is embedded in the embedding.

To explain the distillation of $c$, let us suppose the category of $c$ is color. When the phrase "[*] color" passes through the transformer, [*] and the token "color" would attend to each other. Here, the embedding of the "color" encodes features associated with color from [*]. This can be observed in Fig. 5, which shows changes in the "color" embedding depending on the word in place of [*]. If a word such as "red," "green," or "blue" is in place of [*], the color that the word represents is reflected in the embedding. On the other hand, the embedding does not change if the word does not have any meaning as a color (e.g., circular, stretching, or aerial). Here, putting "[*] color" through the transformer can be viewed as distilling color features from [*].

Based on the observation, we put "[*] [category]" through the transformer in the text encoder, where [category] is a coarse category descriptor of $c$. Then, the forwarded embedding of [category], which we denote as $\mathbf{h}_{[\text{category}]\leftarrow*}$, is used in conjunction with our prompt $y_{\text{custom}}$ as our distilled embedding. The resulting optimization loss is as follows:

$$\mathbb{E}_{\epsilon\sim\mathcal{N}(\mathbf{0},\mathbf{I}),t}\|x_\theta(\mathbf{x}_t, t, \text{Insert}(\tau(\tilde{y}_{\text{custom}}), \mathbf{h}_{[\text{category}]\leftarrow*})) - \mathbf{x}_0\|_2^2. \tag{3}$$

Here, $\tilde{y}_{\text{custom}}$ is a prompt with the [*] token removed from $y_{\text{custom}}$, and Insert() is a function that inserts $\mathbf{h}_{[\text{category}]\leftarrow*}$ back into the encoded $\tilde{y}_{\text{custom}}$.

### 4.3 Stabilizing the optimization

Using the distilled embedding $\mathbf{h}_{[\text{category}]\leftarrow*}$ in conjunction with the detailed training prompt $y_{\text{custom}}$ makes it possible to selectively represent $c$ among all the attributes in the reference image $\mathbf{x}_0$. However, Eq. (3) still forces $\mathbf{h}_{[\text{category}]\leftarrow*}$ to reconstruct all the attributes undescribed in $y_{\text{custom}}$. This conflict between the structure of the embedding and the loss could destabilize the training, making the embedding move to an unexpected direction.

To avoid the conflict and stabilize the training, we propose another learnable embedding called the residual embedding. The purpose of this embedding, which we denote as $\mathbf{h}_{\text{residual}}$, is to capture all the residual attributes except for $c$. Similar to $\mathbf{h}_{[\text{category}]\leftarrow*}$, $\mathbf{h}_{\text{residual}}$ is the forwarded embedding of [category] when the phrase "[R] [category]" passes through the transformer, where [R] is the placeholder token. We set the token "image" as [category] since we do not want $\mathbf{h}_{\text{residual}}$ to be bound to a specific category. Using $\mathbf{h}_{\text{residual}}$ during the training alleviates the need for $\mathbf{h}_{[\text{category}]\leftarrow*}$ to represent all the undescribed attributes, thus stabilizing the training.

One thing to note is that $\mathbf{h}_{\text{residual}}$ could represent $c$ instead of $\mathbf{h}_{[\text{category}]\leftarrow*}$ since it is able to capture attributes of any category. To prevent this, we adopt the cosine similarity loss:

$$\mathcal{L}_{\text{cosine}} = \max\left(0, \frac{\mathbf{h}_{\text{residual}} \cdot \mathbf{h}_{[\text{category}]\leftarrow*}}{\|\mathbf{h}_{\text{residual}}\|\,\|\mathbf{h}_{[\text{category}]\leftarrow*}\|}\right). \tag{4}$$

The loss only updates $\mathbf{h}_{\text{residual}}$, meaning that the residual embedding is forced to move away from what $\mathbf{h}_{[\text{category}]\leftarrow*}$ is representing, which is $c$. Consequently, our final loss is as follows:

$$\mathcal{L}_{\text{recon}} = \mathbb{E}_{\epsilon\sim\mathcal{N}(\mathbf{0},\mathbf{I}),t}\|x_\theta(\mathbf{x}_t, t, \text{Insert}(\text{Prepend}(\tau(\tilde{y}_{\text{custom}}), \mathbf{h}_{\text{residual}}), \mathbf{h}_{[\text{category}]\leftarrow*})) - \mathbf{x}_0\|_2^2, \tag{5}$$

$$\mathcal{L}_{\text{total}} = \mathcal{L}_{\text{recon}} + \lambda\mathcal{L}_{\text{cosine}}, \tag{6}$$

where Prepend() is a function that inserts $\mathbf{h}_{\text{residual}}$ in front of $\tau(\tilde{y}_{\text{custom}})$, and $\lambda$ is a coefficient.

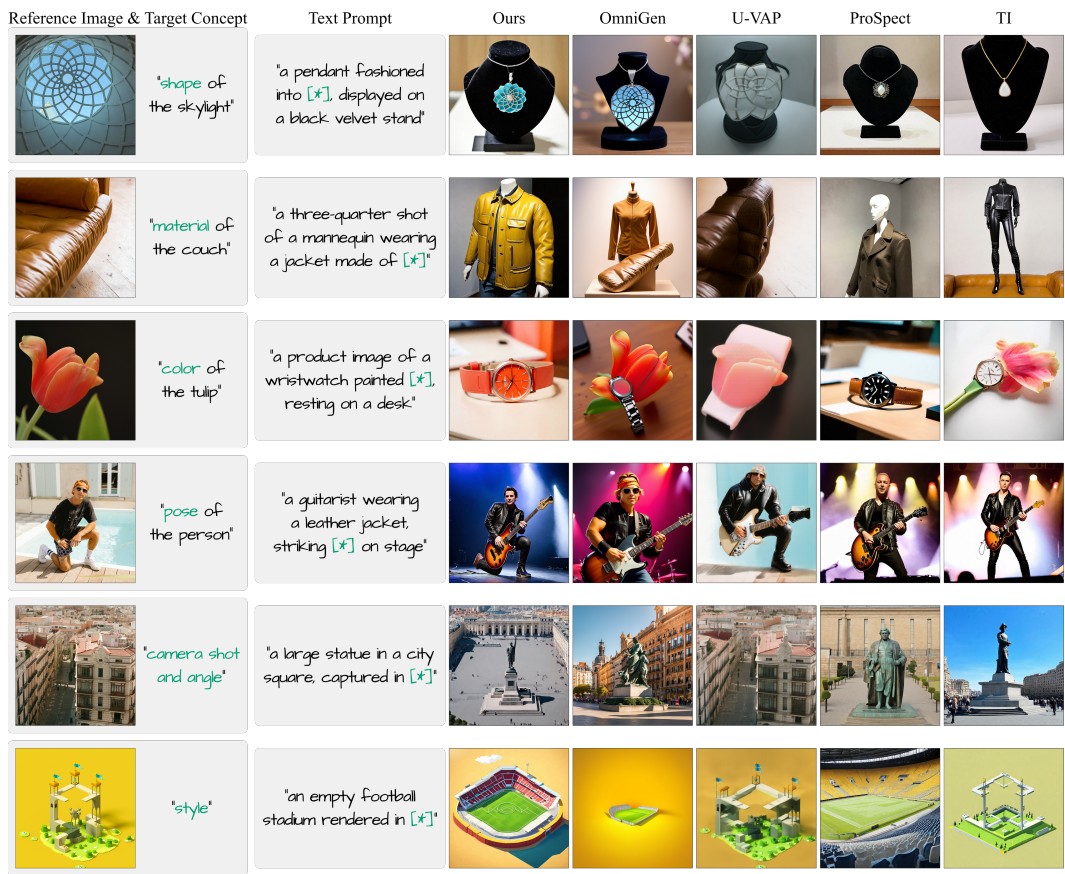

Figure 6: Qualitative comparison with OmniGen, U-VAP, ProSpect, and Textual Inversion (TI).

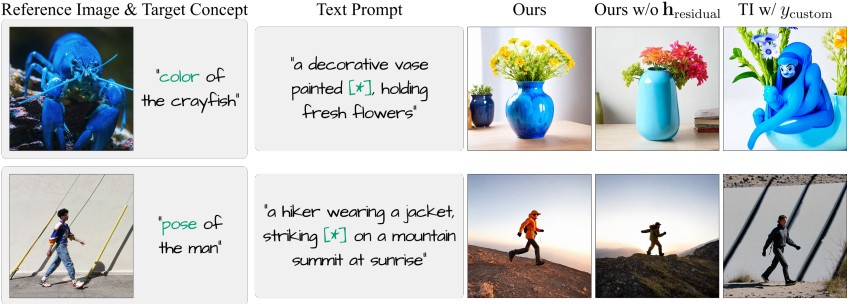

Figure 7: Comparison with ablation setups. Ours is best at exclusively learning target concepts.

## 5 EXPERIMENTS

In this section, we present generation results across diverse target concepts and text prompts using our concept learning method. The concepts and prompts are from our novel dataset that is specifically constructed for the concept learning task. We also provide results from existing methods and ablation setups for comparisons. We discuss the qualitative and quantitative findings in detail below.

**Implementation details.** We have implemented our method on Stable Diffusion 3 (SD3) (AI, 2024) by optimizing token embeddings in each of the three text encoders in the model. In practice, we optimize four token embeddings per encoder. Also, when we omit the [*] token from $y_{\text{custom}}$ to obtain the distilled embedding, we maintain the sentence structure by putting a dummy token in its

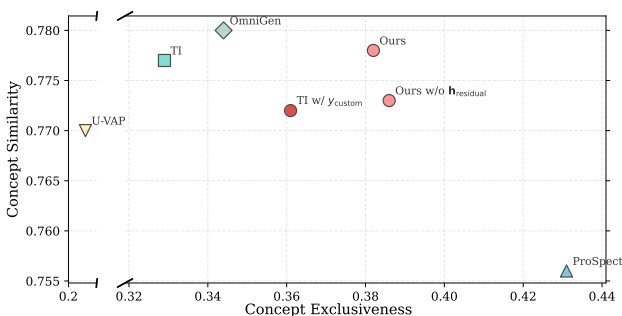

Figure 8: Quantitative comparison with baselines and ablation setups.

place. We use [category] as the dummy token, and its embedding is later replaced with the obtained distilled embedding. Similarly, we insert another dummy token (i.e., "image") at the beginning of $y_{custom}$ and later replace its embedding with the residual embedding.

We conduct all optimization on an NVIDIA GeForce RTX 3090 GPU. We use a learning rate of 0.001 and a batch size of 1. Similar to Textual Inversion, each optimization run takes 5000 iterations on average. The coefficient $\lambda$ from the loss is set as 0.01 or 0.001.

**Proposed dataset.** We construct a dataset dedicated to our attribute-level concept learning task. The target concepts in our dataset are grouped into six categories: shape, material, color, pose, camera shot and angle, and style. These categories correspond to the core visual attributes that make up the composition of an image. We collect real-world images and a set of evaluation prompts for each category, totaling 30 reference images and 60 evaluation prompts. The resolution of the reference images is $1024 \times 1024$, and we carefully select images that exhibit unique attributes of the corresponding category, especially those that are difficult to articulate through text. For the evaluation prompts, we design the prompts of sufficiently diverse contexts that could visually highlight the attributes of the corresponding category. The prompt design is done with the support of a large language model (OpenAI, 2022). We show all the reference images and evaluation prompts in the supplementary material. Our dataset enables a comprehensive assessment of a concept learning method's ability to extract a wide range of attribute-level concepts from an image and apply them across diverse contexts.

**Comparisons.** We compare our method against four baselines: OmniGen (Xiao et al., 2024), U-VAP (Wu et al., 2024), ProSpect (Zhang et al., 2023b), and Textual Inversion (TI) (Gal et al., 2022). All of them have released official source codes. For OmniGen, we use the official implementation as-is. For the other baselines, we modify their code to work with Stable Diffusion 3 before generating results.

As shown in Fig. 6, most baselines either fail to accurately reflect the target concept in the generated image or inadvertently incorporate non-target attributes. This issue is particularly pronounced in U-VAP and ProSpect: U-VAP tends to produce results that closely resemble the reference image, almost reconstructing it, while ProSpect often fails to incorporate the reference concept at all. In contrast, our method reliably integrates the target concept into the output while demonstrating a strong ability to selectively extract only the target concept without bringing in unrelated attributes.

For quantitative evaluation, we generate six images per evaluation prompt, resulting in 60 images per subject and 1,800 images in total. We assess the results using two evaluation metrics, as reported in Fig. 8. Concept Similarity measures how well the generated images reflect the target concept by computing the CLIP similarity between the generation and the reference image. To isolate the relevant concept in both the generated and reference images, we apply concept-specific preprocessing: edge detection for shape and pose concepts, and background removal for material and color concepts. Due to the lack of preprocessing techniques for camera shot and style, we compute the metric for these four concept types. Concept Exclusiveness, on the other hand, measures how well the method avoids embedding irrelevant attributes. It is computed as one minus the CLIP similarity between the unprocessed generated image and the reference image, and a high score implies that fewer non-target attributes have been copied over. As shown in Fig. 8, our method achieves the highest overall scores in concept similarity and exclusiveness, indicating its effectiveness in accurately

| Metric | OmniGen | U-VAP | ProSpect | TI |
|---|---|---|---|---|
| **Concept Similarity (%)** | **58.1** | **54.2** | 94.7 | **68.8** |
| **Concept Exclusiveness (%)** | **51.2** | **79.2** | 27.4 | **57.1** |

| Metric | Ours | OmniGen | U-VAP | ProSpect | TI |
|---|---|---|---|---|---|
| **Prompt Fidelity** | **0.842** | 0.791 | 0.181 | 0.811 | 0.597 |

Table 1: User study results.

capturing and isolating the target concept. While ProSpect scores higher in concept exclusiveness, the method largely ignores the target concept and generates images based solely on the text prompt, resulting in low similarity to the reference image regardless of the target concept.

We also conduct a user study involving 11 participants, each responding to 30 questions, totaling 330 responses. Each question presents a reference image, a target concept, a text prompt, and two generated images—one from our method and one from a baseline. Participants are asked three questions: (1) Which image better reflects the target concept? (concept similarity), (2) Which image appears less copied from the reference image? (concept exclusiveness), (3) Does the image faithfully follow the prompt or not? (prompt fidelity; evaluated per image). As shown in Table 1, our method outperforms the baselines in both concept similarity and concept exclusiveness in most cases. The only exception is against ProSpect in concept exclusiveness, where it scores higher. Nevertheless, ProSpect still underperforms in prompt fidelity, and our method achieves the highest prompt fidelity score among all methods.

**Ablation studies.** We also compare our method with several ablation setups to assess the contribution of each component. As shown in Fig. 7, removing the residual embeddings leads to unstable training and results in distilled embeddings that fail to capture the target concept effectively. In another variant where only custom training prompts are used without other components, we observe that non-target attributes are not fully disentangled from the target concept, leading to their unintended inclusion in the generated images. Also, our method achieves the highest overall scores in concept similarity and exclusiveness in Fig. 8. These results highlight the importance of each component in our method, and demonstrate that all elements are necessary to achieve accurate and controlled concept learning.

## 6 LIMITATIONS AND DISCUSSION

We conducted the experiments using SD3 due to limited GPU resources, and the generation quality of our method is constrained by that of the base model. For example, SD3 is known to have a limited understanding of human anatomy, which can reduce the accuracy in extracting complex human poses from reference images (Edwards, 2024). Also, the range of contexts in which the extracted concepts can be applied is limited by the generative capabilities of SD3. Fortunately, our method does not depend on SD3, but improving such capabilities still requires adopting a larger model (e.g., FLUX (Labs, 2024)).

In addition, our experiments are designed to span six distinct categories of visual concepts, each selected to test different aspects of compositional and controllable generation. We keep the number of reference images within a manageable range, as our goal is to evaluate whether a method can extract, isolate, and generalize a concept across diverse contexts. Accordingly, we prioritize diversity across concept types and contexts, resulting in 1800 generations per method and fine-grained comparisons across multiple criteria. Moving forward, we encourage future work to expand in complementary directions, such as developing scalable tools for evaluation in terms of both difficulty and volume.

## 7 CONCLUSION

We present a novel method for learning diverse concepts from a single reference image by isolating the target attribute. Our distilled and residual embeddings enable precise concept extraction and stable optimization. Experiments show that our approach outperforms existing methods, offering a flexible solution for attribute-level manipulation in generative models.

ETHICS STATEMENT

This work aims to assist designers in efficiently incorporating specific visual attributes into the image generation process, which could enhance creativity and reduce prototyping costs in various domains. However, as with many generative models, the potential for misuse remains. There is a risk of generating images that unintentionally reflect or reinforce social biases or violate copyright if used improperly. We encourage future work to investigate safeguards and bias-mitigation strategies when deploying such tools in real-world applications.

USE OF LARGE LANGUAGE MODELS

We used large language models solely to aid or polish writing. They were not used for idea generation, technical contributions, experiments, or analysis.

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
