# OpenReview forum: "Selectively Extracting and Injecting Visual Attributes into Text-to-Image Models"
_ICLR.cc/2026/Conference — ICLR 2026 Conference Withdrawn Submission_

### Official Review · Reviewer_Rkez · 2025-10-25

**Soundness:** 3
**Presentation:** 2
**Contribution:** 2
**Rating:** 4
**Confidence:** 4

**Summary:**

The paper presents a method for learning specific visual attributes (e.g., shape, color, material) from a given input image. Once such a visual attribute is learned, the user can generate new images depicting objects that share this attribute.
The method learns a new textual token embedding that represents the desired visual attribute. To ensure that the learned token captures only the intended attribute, the method incorporates three mechanisms:
First, the sentence used during optimization explicitly describes the other visual attributes present in the image. Consequently, the learned token does not need to encode those attributes in order to reconstruct the image.
Second, instead of directly using the learned token in the prompt, a distilled token is used. This distilled token represents the category of the visual attribute and is influenced by the learned token during optimization. Since it captures only category-relevant information, it filters out unrelated details.
Third, an additional token is optimized alongside the token representing the learned visual attribute, and a loss term is applied to enforce orthogonality between them, encouraging the tokens to encode distinct information.

**Strengths:**

- The task is well-motivated.
- The visual results that are provided are plausible. The method allows to learn a specific visual attribute from an image by specifying it with text.
- The examples are diverse, demonstrating that the method can be applied to diverse visual attributes.

**Weaknesses:**

The method requires a lengthy optimization process for each image, whereas other approaches addressing the same task operate in a feed-forward manner [1,2].
- The paper does not demonstrate compositional capabilities, which have been shown in prior works [1,2,3].

These two points make the scope of the paper relatively limited compared with existing methods. Furthermore, the proposed approach is not compared against these works, so it remains unclear whether it outperforms them, even without considering composition.

- The technical novelty appears somewhat incremental. Similar (though not identical) techniques have been used in prior works as a form of regularization for learning specific attributes. For example, in [4], a negative token was optimized with a similar purpose to the orthogonal token.

- It would be helpful to include more examples in Figure 4. Without the distilled embedding, it seems that the token learns the concept of rainbow. Do you have an explanation for this behavior?
- In Figure 5, it is unclear which prompts were used to generate each image (do the prompts contain only the specified token?). It is also not clear whether the seed was fixed, and why the images generated from the distilled token appear so noisy.


In summary, my main concern lies in the paper’s limited scope. Combined with the relatively modest novelty, I lean toward rejection.


[1] Dorfman, Sara, et al. "Ip-composer: Semantic composition of visual concepts." Proceedings of the Special Interest Group on Computer Graphics and Interactive Techniques Conference Conference Papers. 2025.

[2] Lee, Sharon, et al. "Language-Informed Visual Concept Learning." The Twelfth International Conference on Learning Representations.

[3] Garibi, Daniel, et al. "Tokenverse: Versatile multi-concept personalization in token modulation space." ACM Transactions on Graphics (TOG) 44.4 (2025): 1-11.

[4] Dong, Ziyi, Pengxu Wei, and Liang Lin. "Dreamartist: Towards controllable one-shot text-to-image generation via positive-negative prompt-tuning." arXiv preprint arXiv:2211.11337 (2022).

**Questions:**

See Weaknesses section.

---

### Official Review · Reviewer_JXQj · 2025-10-28

**Soundness:** 2
**Presentation:** 3
**Contribution:** 2
**Rating:** 4
**Confidence:** 4

**Summary:**

This paper tackles the challenge of selective, attribute-level concept learning in text-to-image generation. Visual attributes from a single reference image are extracted and injected the into the generation of new images. It addresses the challenge of accurate extraction, where only the selected concept is used for the T2I conditioning without dragging along entangled features. The method learns two token embeddings from the reference image: the distilled embedding isolates the chosen attribute using attention over the phrase [*][category], while a residual embedding that captures everything else. A VLM-crafted custom prompt describes non-target content, so the distilled embedding isn’t forced to encode it, using cosine loss to keep the residual orthogonal to the distilled. The main contribution of this work is a dataset-free method for accurate attribute control in text-to-image generation.

**Strengths:**

1. The applicability is given as there is no heavy training or cumbersome setup of the method involved and it is intuitive to use.
2. Efficiency, as there is no dataset necessary and it essentially enables one-shot learning of a specific concept to guide t2i
3. It is addressing the well-known gap of specific attribute control in text-to-image generation. Existing methods often only enable high-level, unspecific or inaccurate control of concepts in image generation.
4. The methodological foundations are sound and the qualitative results look decent.

**Weaknesses:**

I assess the current paper as marginally below acceptance threshold. I find the idea promising and practically impactful, but it is missing solid evidence to support the claimed contribution. My concerns are mostly directed towards the quality and accuracy of the attribute control. Please refer to the below list of Weaknesses and questions. Additional evaluations on concept generalizability, the granularity of the control,  image-editing applicability, and image fidelity would significantly strengthen the contribution.

1. The authors show that it works on 6 selected concepts but leave it unclear how broadly applicable this is. Can the concept also be a more complex or more detailed task? Is it possible to provide more example concepts?

2. Text-to-image generation is well-known, but modifying concepts in existing images is what is practically equally as useful, if not more. For an impactful contribution, I would like to see evidence that this method also be used to control selected attributes in existing images as this would support the real-world applicability of the method.

3. How fine-grained/detailed control does this mechanism allow? Would it be possible to e.g. control only the pose of the arm of a human or the color of only a part of an object? This corresponds to the expressiveness of the proposed embeddings and if the attention mapping can capture fine-details.

4. Does it also work in scenarios, where there is no single prominent object in the image, e.g. controlling the pose of a group of people or the color of an object when many more objects are present.

5. Does using the method impact the quality/fidelity of the generated images? How is the quality overall and is better or worse compared to other methods? There is no quantitative evaluation on overall image quality and the qualitative results give me a mixed impression.

6. Is the approach model agnostic and therefore applicable to other image generation models? The authors note that it should be scalable to models like FLUX, but present no examples/evidence of for this claim. Applicability to other T2I-frameworks would significantly support the relevance of the contribution as an adaptable framework rather than an optimized solution for SD3.

**Questions:**

1. Figure 4 is difficult to understand as a direct comparison. Are the left images in a) and b) visualizations of the embeddings? Please clarify in more detail what is shown here.
2. In chapter 5 it is mentioned that the alternative methods are modified to work with Stable Diffusion 3. How is it ensured that methods not developed for SD3 unleash their maximum capacity? Using the proposed method on SD2 could be a good comparison to underline its superiority compared to alternatives.
3. Table 1 is difficult to understand. I assume the percentage score indicate the win percentage of the proposed method against the shown alternatives? Please clarify this or find another format for the table that makes this clear.

---

### Official Review · Reviewer_VvW3 · 2025-10-31

**Soundness:** 3
**Presentation:** 3
**Contribution:** 3
**Rating:** 6
**Confidence:** 4

**Summary:**

This paper proposes a method to extract and inject attribute-level visual concepts from a single reference image into text-to-image (T2I) diffusion models. The approach enables controllable and disentangled concept transfer across diverse prompts, and demonstrates performance on par or superior to existing baselines.

**Strengths:**

- The proposed method is well-motivated, and each component is designed with a clear and intuitive rationale.
- Despite its simplicity, the approach appears versatile and could be easily applied to various Textual Inversion (TI) variants.
- The paper shows solid empirical results, achieving competitive or superior performance compared to strong baselines.

**Weaknesses:**

- Although this paper demonstrates solid overall performance, some of the evaluations are insufficient to fully support its claims. For instance, the user study shows that the performance gap between the proposed method and OmniGen is not substantial. In fact, I feel that the results appear quite similar. The paper should provide a deeper analysis or discussion explaining why this margin is small. Furthermore, the user study includes only 11 participants, which raises concerns about the statistical significance and reliability of the findings.
- The paper provides only two ablation experiments. A more in-depth analysis is required and it will enhance the credibility of the model. For example, examining how sensitive the performance is to the placement of residual or distilled tokens within the architecture, exploring hyperparameter sensitivity, and so on.

**Questions:**

- Given its general formulation, the proposed method could potentially be extended beyond TI. It would be interesting to discuss or test how it performs with other TI variants such as DreamStyler or P+.

---

### Official Review · Reviewer_9d32 · 2025-11-03

**Soundness:** 3
**Presentation:** 2
**Contribution:** 2
**Rating:** 4
**Confidence:** 3

**Summary:**

The paper introduces a novel method for text-to-image generation that extracts specific visual attributes from a reference image and injects them into the generation process. ​It optimizes a text token to exclusively represent the target attribute using a custom training prompt and distilled/residual embeddings, which select attributes related and unrelated to the target concept. The proposed method outperforms existing approaches such as basic text inversion.​

**Strengths:**

+ allows more fine-grained reference image attribute control for image generation
+ simple prompt optimization based approach without need of training
+ enables new applications

**Weaknesses:**

- idea is incremental as regarding to textual inversion
- limited experimental results, only 4 categories are show, not sure about generalization to other concept types
- comparison with general reference image condition is missing, such as control net with edge and pose condition, which can also extract certain types of attributes from reference image.

**Questions:**

what are the cases the proposed method cannot work well

---

### Note · Authors · 2025-11-14

I have read and agree with the venue's withdrawal policy on behalf of myself and my co-authors.